# Human Patient-Derived Brain Tumor Models to Recapitulate Ependymoma Tumor Vasculature

**DOI:** 10.3390/bioengineering10070840

**Published:** 2023-07-15

**Authors:** Min D. Tang-Schomer, Markus J. Bookland, Jack E. Sargent, Taylor N. Jackvony

**Affiliations:** 1UConn Health, Department of Pediatrics, 263 Farmington Avenue, Farmington, CT 06030, USA; taylornjackvony@gmail.com; 2Connecticut Children’s Medical Center, 282 Washington St., Hartford, CT 06106, USA; mbookland@connecticutchildrens.org; 3The Jackson Laboratory for Genomic Medicine, 10 Discovery Drive, Farmington, CT 06030, USA; john.e.sargent@gmail.com

**Keywords:** brain tumor model, tumor microenvironment, transcriptomics, cancer stem cells, tumor spheroid, neurosphere, homeobox

## Abstract

Despite in vivo malignancy, ependymoma lacks cell culture models, thus limiting therapy development. Here, we used a tunable three-dimensional (3D) culture system to approximate the ependymoma microenvironment for recapitulating a patient’s tumor in vitro. Our data showed that the inclusion of VEGF in serum-free, mixed neural and endothelial cell culture media supported the in vitro growth of all four ependymoma patient samples. The growth was driven by Nestin and Ki67 double-positive cells in a putative cancer stem cell niche, which was manifested as rosette-looking clusters in 2D and spheroids in 3D. The effects of extracellular matrix (ECM) such as collagen or Matrigel superseded that of the media conditions, with Matrigel resulting in the greater enrichment of Nestin-positive cells. When mixed with endothelial cells, the 3D co-culture models developed capillary networks resembling the in vivo ependymoma vasculature. The transcriptomic analysis of two patient cases demonstrated the separation of in vitro cultures by individual patients, with one patient’s culture samples closely clustered with the primary tumor tissue. While VEGF was found to be necessary for preserving the transcriptomic features of in vitro cultures, the presence of endothelial cells shifted the gene’s expression patterns, especially genes associated with ECM remodeling. The homeobox genes were mostly affected in the 3D in vitro models compared to the primary tumor tissue and between different 3D formats. These findings provide a basis for understanding the ependymoma microenvironment and enabling the further development of patient-derived in vitro ependymoma models for personalized medicine.

## 1. Introduction

Ependymoma is the third most common brain tumor in children and is incurable in more than half of cases [1]. Ependymomas are glial tumors that arise throughout the neuroaxis, including the supratentorial brain, which comprises the cerebral hemispheres with the posterior fossa encompassing the cerebellum and brainstem and in the spinal cord [2]. The posterior fossa tumors occur more commonly in children, while supratentorial and spinal tumors dominate in adults [3]. It presents considerable histopathological variation and biological heterogeneity [4]. The total resection and spinal location represent the best-studied positive clinical predictors of the outcome [5,6]; however, the complete removal of the intracranial ependymoma is not always possible and results in poor outcomes [2]. Surgery and radiation therapy are the main therapeutic strategies, whereas the role of chemotherapy for intracranial ependymoma remains uncertain, and ependymomas are prone to chemoresistance [7,8]. Thus, there is an urgent need for novel therapeutic targets and options. Towards this end, genomic profiling studies have recently identified significant genetic and epigenetic drivers of ependymoma [9,10,11,12,13,14,15,16]. C11orf95–RELA fusions have been observed to occur in two-thirds of pediatric cases of supratentorial ependymomas and are believed to be oncogenic due to increased NF-kB signaling [17]. Furthermore, a subtype of cerebellar ependymomas that is associated with young patient age and poor prognosis is characterized by a CpG island methylator phenotype (CIMP) and Polycomb repressive complex 2, which is driven by the trimethylation of H3K27 [18]. Testing therapies against these molecular targets requires preclinical models that are representative of different ependymoma molecular types; however, such models are lacking, and ependymoma cell lines are absent.

Human cell culture-based models provide an alternative option for the superior affordability, flexibility, and scalability of in vitro systems compared to animal models. However, the in vitro culturing of primary pediatric ependymoma has so far failed to produce cell lines [19]. Patient-derived xenograft mouse models have become indispensable tools for cancer studies and drug development. Numerous reports have shown that the implantation of patient brain tumor cells into matching locations of immune-deficient mice closely replicated the histopathology, invasive growth, cancer stem cell pools, and key molecular genetic abnormalities of original patient tumors [20]. However, the engraftment rate varied greatly and was <50% for most brain tumor types [21]. Table 1 summarizes the past attempts at human ependymoma modeling with both cell cultures and animal models. Cell culture approaches of the 2D monolayer or 3D neurosphere/spheroid have largely been unsuccessful or, at most, inconsistent in preserving ependymoma in vivo characteristics in the long term. Recent efforts in the patient-derived orthotopic xenograft model of ependymoma have reported the in vivo loss of tumorigenicity, highlighting challenges to this approach [22].

We have developed bioengineered 3D brain tissue models that can be adapted for human patient brain tumor tissue [38]. This system was originally designed to support normal brain tissue, as demonstrated with rodent brain cells [38,39] and human ‘normal’ brain cells from epilepsy surgery [40]. To adapt the 3D brain tissue model for brain tumors, this system provided several useful properties: The silk scaffold has mechanical properties that are similar to the cortex [38]. The silk fibroin material is inert without biological interference with tumor cells yet al.lows for cell attachment with an appropriate surface coating such as polylysine for neural cells. The porous structure contains the ECM gel in a stable 3D architecture that prevents the degradation of hydrogels and subsequent structural collapse, which has been known to negatively impact 3D tissue growth in scaffold-free hydrogel systems. The composite design also allows for the independent examination of bio-active components, such as ECM types [35,41], and from the biophysical properties of a 3D structure, such as stiffness and shapes [38]. Finally, the donut-shaped compartmentalized design allowed the separation of the scaffold region and the ECM region for different cell types and/or interactions. Most recently, we extended the 3D brain tissue model to major types of brain tumors [42] by defining the starting condition of 3D models for different brain tumor types as encountered by the pediatric population.

Ependymoma presents a unique pseudo-rosette pattern of tumor cells surrounding capillary vessels [43]. This vasculature was found to play a critical role in ependymoma aggressiveness and patient prognosis [44]. To recapitulate the tumor vasculature, a 3D architecture is necessary for cell–cell and cell-ECM signaling and interactions. In this study, we aimed to reconstitute the tumor vasculature of intracranial ependymoma, given the clinical relevance of VEGF signaling and anti-VEGF treatment for ependymoma [45,46]. The comparative gene expression analysis of the 3D co-culture model with the primary tumor tissue as well as other model formats could also illuminate the regulatory roles of different microenvironmental factors, thus guiding future model optimization efforts. 

## 2. Materials and Methods

### 2.1. Patient Brain Tumors

Human patient brain tissue was obtained from epilepsy neurosurgery at Connecticut Children’s Medical Center (CCMC) in Hartford, Connecticut. The procedures were approved by the Institutional Research Boards of UConn Health Center and CCMC (IRB #13-035). Informed consent was obtained from all human patients prior to the surgery. All methods were performed in accordance with the guidelines and regulations of the approved IRB protocol. The tissue specimen was transported in a chilled RPMI-1640 medium (Sigma-Aldrich, St. Louis, MO, USA) containing 1% penicillin-streptomycin (Pen/Strep, Thermo Fisher, Waltham, MA, USA) and 5% fetal bovine serum (FBS) on an ice pack from the operation room to the laboratory in <4 h post-surgery. 

### 2.2. Brain Tumor Tissue Dissociation 

The tissue specimen was weighed, cut into ~1 mm^3^ pieces with a sterile razor blade, and re-suspended at 1600 mg tissue/10 mL in Hibernate-A medium (Thermo Fisher) containing 1% Pen/Strep and primocin (10 µg/mL, InvivoGen, San Diego, CA, USA). The tissue suspension was treated with a cocktail of enzymes (DNase I, 50 U; dis-pase II, 5 U; collagenase I, 1 U and collagenase IV, 10 mg/mL in 10 mL 0.5% Tryp-sin-EDTA solution) at 37 °C for 20 min, followed by neutralization with a 10 mL trypsin inhibitor (0.5%, *w*/*v*) solution and gentle pipetting. The tissue dissociation solution was filtered with a 100-µm cell strainer (Fisher Scientific, Suwannee, GA, USA), and single cell suspension was collected. 

### 2.3. 3D Silk Protein-Based Scaffolds and Extracellular Matrix (ECM) Gel Preparation

Silk solution and porous scaffolds were prepared from Bombyx mori cocoons as described previously [39]. Salt-leached porous silk mats 100 mm in diameter were provided by David Kaplan’s laboratory at Tufts University. A biopsy punch was used to generate a donut-shaped silk protein-based scaffold (outer diameter, 5 mm; inner diameter, 2 mm; height, 2 mm). Silk scaffolds were autoclaved, coated with poly-L-lysine (10 µg/mL, Sigma) overnight, and washed 3 times with phosphate-buffered saline (PBS, Sigma). Collagen gel was prepared from a high-concentration rat tail type I collagen (8–10 mg/mL, Fisher Scientific), 10X M199 medium (Thermo Fisher), and 1 M sodium hydroxide mixed at a ratio of 88:10:2, followed by gelling at 37 °C for 1–2 h. Matrigel (~10 mg/mL, growth factor reduced, Fisher Scientific) was mixed in a 1:1 ratio with a collagen gel solution (8–10 mg/mL) before infusing the silk scaffolds. To make a scaffold–gel composite structure, the cell-laden scaffolds were transferred to a new dish, dabbed on the dry surface for a few times to deplete the free-flowing liquid, and then infused with the liquid ECM gel. The scaffold/gel composite was incubated at 37 °C for 1 h for the gel to solidify before culture medium immersion. 

### 2.4. Cell Seeding in 2D and 3D

For 2D cultures, cells were plated at 250,000 cells/well in 6-well plates. For 3D scaffold-based cultures, the scaffolds were immersed in high-density cell suspensions (~100 million cells/mL) for 24 h, followed by extensive washes with media, and proceeded to scaffold-only cultures or ECM gel-infused composite cultures. The culture media used: NeuralBasal/B27 (Invitrogen, Grand Island, NY, USA) supplemented with a 20 ng/mL recombinant human fibroblast growth factor, basic-154 (FGF, ConnStem, Cheshire, CT, USA) and a 20 ng/mL human epidermal growth factor (EGF, PeproTech, Rocky Hill, NJ, USA), termed “N” medium; NeuralBasal/B27 with EGF and FGF and supplemented with 10% fetal bovine serum (FBS, Denville Scientific, Metuchen, NJ, USA) (“N+FBS”), and an “N” medium mixed at 1:1 with endothelial growth media EGM-2MV (Lonza, Walkersville, MD, USA) without serum (“E”), termed the “N+E” medium. In some cases, the endothelial growth factor VEGF was omitted from the combined media, termed “N+E (no V)”. DMEM (Invitrogen) media without or with the boluses for the “E” (“DMEM+Supp.”) were also used for comparison. Media were changed once a week for all culture systems.

In some cases, “Sphere” cultures were developed in ultra-low cell attachment 96-well plates, by plating 10,000 tumor cells per well and culturing the cells as a suspension. 

### 2.5. Endothelial Cell 3D Co-Culture

Human dermal microvascular endothelial cells (hMEC/D3, MilliporeSigma, Berlington, MA, USA) were used. To introduce endothelial cells, a tumor cell-laden 3D-SF model was removed from its well, transferred to a new dish, and dabbed on the dry surface a few times to deplete free-flowing liquid. A cell pellet containing 10,000 endothelial cells was mixed into the Matrigel (made with a 1:1 ratio with collagen as described above) and injected into the center hole region of the donut-shaped 3D model. The gel was solidified at 37 °C for 1 h in the incubator before culture medium (N+E) immersion. 

### 2.6. Tissue Viability Assay

The AlamarBlue assay was used to measure the cell viability of 3D cultures according to the manufacturer’s protocol (ThermoFisher Scientific). Briefly, the alamarBlue reagent was mixed in fresh culture media (1:10, *v*/*v*) and incubated for 2 h at 37 °C. The solution was transferred into a new 96-well plate. The fluorescence intensity was read at Ex./Em. of 560/590 nm on a micro-plate spectrophotometer (SynergyMx Gen5, BioTek, Winooski, VT, USA). Four replicate cultures per group per time-point were used for this assay, and the readings were normalized against the media controls. 

### 2.7. Flow Cytometry Cell Counting

Cells were treated with 0.5% trypsin-EDTA (5 min) (Invitrogen) or 0.25% trypsin-EDTA (25 min) for 2D and 3D cultures, respectively. Cell suspensions were mixed at 1:1 with medium containing 10% FBS and were centrifuged at 300× *g* for 5 min. Cell pellets were re-suspended in PBS containing 2% FBS and stained on ice for 15 min with eFluor 780 (Affymetrix eBioscience, San Diego, CA, USA). The cells were washed in 2% FBS-containing PBS by centrifuging at 300× *g*, 5 min. The cell pellets were re-suspended, stained with membrane-bound flow antibodies on ice for 20 min and washed. Stained/washed cells were fixed with 4% paraformaldehyde (Electron Microscopy Sciences, Hatfield, PA, USA) for 20 min, and washed, and permeabilized with PBS containing 0.1% Tween and 0.2% FBS for 20 min. The cells were subsequently stained with intracellular flow antibodies for 30 min, washed, and proceeded to flow cytometry. The flow antibodies used were anti-Ki67-eFluor 450 (eBioscience) and anti-Nestin-Alexa 647 (Biolegend, San Diego, CA, USA). 

Flow cytometry was performed on a BD LSR II instrument equipped with 5 lasers with BD FACS DIVA software (BD Biosciences, San Jose, CA, USA). In total, 2000–5000 cells per sample were counted and analyzed with FlowJo software (Flow-Jo, Ashland, OR, USA). Unstained cells were used to set a gate for “live & single” cells. eFluor 780-stained cells were used to set the gate for “live” cells and “control” gates for each stain with a threshold of 0.5% (i.e., <0.5% cells were positive for the respective stain). The positive cell population corresponding to a stain was calculated from the multiplexed cell population using the same gate as that used for the control unstained cell population.

### 2.8. Immunofluorescence Staining and Imaging

Cell cultures were fixed with 4% paraformaldehyde (Electron Microscopy Sciences) for 20 min, washed, and permeabilized with PBS containing 0.1% Triton X-100 (Fisher Scientific) and 4% normal goat serum (Jackson ImmunoResearch Labs, West Grove, PA, USA) for 20 min, followed by the incubation of primary antibodies overnight at 4 °C. After three 10 min PBS washes; the cells were incubated with secondary antibodies for 1 h at room temperature followed by extensive washes. The antibodies included: anti-Nestin (mouse clone 10C2, 1:100, eBioscience), anti-Ki67 (mouse clone B56, 1:100, BD Biosciences), anti-glial fibrillary acidic protein (GFAP, mouse clone GA5, 1:500, eBioscience), anti-Vimentin (mouse clone RV202, 1:200, BD Bioscience), anti-alpha smooth muscle actin (SmA, rabbit clone E184, 1:100, Abcam, Cambridge, MA USA), and anti-PECAM (rabbit, 1:100, Abcam). Goat anti-mouse or rabbit Alexa 488 and 568 (1:250; Invitrogen) secondary antibodies were used. Fluorescence images were acquired on a Leica DM IL fluorescence microscope using an excitation/emission (Ex/Em) of 470/525 nm for Alexa 488 and Ex/Em of 560/645 nm for Alexa 568. Confocal images were acquired on a Zeiss 780 laser scanning confocal imaging system. 

### 2.9. RNA-Seq 

RNAs were extracted with a Qiagene AllPrep kit on a QiaCube automated station. The samples were sequenced by JAX-GM Genome Technologies Core. RNA-seq libraries were prepared with a KAPA Stranded mRNA-Seq kit. The quantification of libraries was performed using a real-time qPCR. Sequencing was performed on an Illumina Hiseq 4000 platform, generating paired-end reads of 75 bp. Raw reads obtained from the sequencer were processed, including quality control steps to identify and remove low-quality samples. Reads with more than 50% low-quality bases (>Q30) overall were filtered out, and the remaining high-quality reads were then used for expression estimation. 

### 2.10. Differential Gene Expression 

Transcriptome data analysis was performed using R (version 4.3.0) statistical software with rounded gene-level abundance estimates from RSEM [47]. Gene annotations for transcripts were gathered from Ensembl (release 99) [48] using biomaRt (version 2.42.0) [49]. The normalization of read counts, variance stabilizing transformation, and filtering low gene variance with the detection of differential expression was performed using the DESeq2 (version 1.40.0) [50] package (parameters: “alpha = 0.01”). Figures were created using the ggplot2 (version 3.4.2) and ComplexHeatmap (version 2.16.0) [51] packages. An estimated log2 fold change was corrected for the expression level by a shrinkage procedure [52] and plotted using the gene annotation package (org.Hs.eg.db, version 3.17.0) and EnhancedVolcano (version 1.18.0) package [53]. 

### 2.11. Statistical Analysis

Data are the mean ± standard error of the mean (S.E.M.), except where otherwise noted. The analysis used Student’s *t*-test, except for the cell percentage data. For all tests, *p* < 0.05 was considered significant. For the statistical analysis of flow cytometry-measured cell percentages, the construction of simultaneous confidence intervals was performed, as we previously described [40]. A program written in R was used to implement the analysis.

## 3. Results

### 3.1. Study Design of the Optimization of Cell Culture Conditions for Intracranial Ependymoma

Figure 1 illustrates the study design for the optimization of ependymoma cell culture conditions. To identify the improved liquid environment, we tested chemically defined media, including the DMEM media, neural stem cell media “N”, or the VEGF-containing endothelial culture media “E” with or without FBS. N and E mixed at a 1:1 ratio were also used (“N+E”). To identify an improved 3D environment, we compared 3D cultures on the silk scaffolds only (“SF-only”) and scaffolds infused with an extracellular matrix such as collagen type I (“Col”) or Matrigel (“Matri”, mixed at 1:1 with collagen type I for enhanced structural stability). Endothelial cells were included in 3D co-culture models. These in vitro models were assessed by the cell/tissue growth parameters including morphology, viability, and cell composition, to determine an optimal combination of the liquid and 3D microenvironments. Finally, these models were compared with a matched patient tumor for transcriptomic profiling to determine their similarities regarding gene expression patterns.

In this report, all four cases were of the posterior ependymoma type (Table 2). We used only primary cultures, i.e., not passaged, for the studies.

### 3.2. Media Conditions Affect Ependymoma Cell Growth and 3D Model Viability 

We used vimentin (Vim) as an ependymoma marker [54] and alpha-smooth muscle actin (SmA) as a fibroblast marker to characterize the media’s effects on cell selection from the primary tumor tissue (Figure 2A). Three-week cultures from the EPN-1 sample showed that the N+E media (Figure 2A(e–h)) resulted in the most cell growth compared to the N (Figure 2A(a–d)) or DMEM media (Figure 2A(i–p)). The presence of FBS resulted in a decrease in spindly-shaped Vim+ tumor cells and an increase in flat-shaped SmA+ fibroblasts in a dose-dependent manner (Figure 2A(i–p)). 

To evaluate 3D model viability, the AlamarBlue assay was used (Figure 2B). The assay result of the 3D SF-only cultures from EPN-1 was consistent with 2D cultures, showing higher tissue viability in the N+E media than in N or N+FBS with significant differences at most time points from 2 wk up to 6 wk (Figure 2B(a)). For EPN-2 (Figure 2B(b)), the added E condition also showed that the N+E media outperformed all other media conditions, including N or E or N+FBS, indicating that the combining of liquid conditions supported both the neuronal and endothelial cell types, which was necessary for ependymoma in vitro growth. 

Because the NeuralBasal base media in N was formulated with hypotonic osmolarity to specifically favor neuronal growth [22] and the E media contained VEGF as an essential growth factor for endothelial cell growth, we tested whether low osmolarity or VEGF was necessary for ependymoma in vitro growth. Therefore, for EPN-4 (Figure 2B(c)), three media conditions were compared: N+E, N+E without VEGF (“noV”), and DMEM with the same supplements as N+E (“+Supp”). The AlamarBlue assay showed that the N+E media yielded significantly higher tissue viability than without VEGF or with DMEM base. 

Together, these data suggest that the N+E media provided an improved liquid environment for ependymoma in vitro growth compared to existing options and that VEGF was necessary for the growth of ependymoma. 

The AlarmaBlue assay results of these ependymoma cases also indicated that the most robust growth occurred between the third to the sixth week in the culture. Therefore, the following studies focus on 3D cultures up to 6 weeks in vitro.

### 3.3. Nestin+ Cell Enrichment in 2D Ependymoma Cultures by the Improved N+E Media 

We used flow cytometry to quantify different cell types in cultures, including neural, stem/progenitor cells (Nestin), cancer stem cells (CD133) neuronal (TUJ1), glial (GFAP), and the proliferative population (Ki67). Among the cell types we tested, the Nestin+ population showed the strongest correlation with media differences of ependymoma cultures (Figure 3). For EPN-1, Nestin+ cells had a higher percentage (72.6%) in N+E than in N (34%) or FBS-containing DMEM, i.e., 37.3% and 36.9% of 2% and 10% serum-containing DMEM, respectively (Figure 3A). For EPN-2, there was a higher percentage of Nestin+ cells in N+E (36.3%) compared to N+FBS (14.2%) or E (5.77%) (Figure 3B). For EPN-4, Nestin+ cell percentages were higher in N+E (70.7% and 56.3% at 1 wk and 3 wk, respectively) compared to N+E without VEGF (32.9% and 37.2% at 1 wk and 3 wk, respectively) (Figure 3C).

### 3.4. Nestin+ Cells Are the Predominant Proliferating Cells during Ependymoma In Vitro Growth 

To determine whether the Nestin+ cells were proliferative, we examined the time course of Nestin+ cell population changes and its co-staining pattern with Ki67 in EPN-3 samples (Figure 4). An analysis of different time points showed that the Nestin+ population showed a dramatic increase in 3D-SF cultures, while this was relatively unchanged in the 2D cultures (Figure 4A). Flow cytometry analysis showed that the Nestin+/Ki67+ population represented 83% of Ki67+ cells (Figure 4B), indicating that the Nest+ cells were the predominant proliferating cells in the culture. Indeed, immunostaining images showed that these two markers were largely co-localized in the same cell population (Figure 4C).

### 3.5. Nestin+/Ki67+ Cells form Tumor Stem Cell Niche during Ependymoma In Vitro Growth

We extended our analysis to the other three ependymoma cases (Figure 5). Flow cytometry analysis showed that the Nestin+/Ki67+ population represented the majority of Ki67+ populations, i.e., of 78.3%, 86.8%, and 91.4% for EPN-1, two, and four samples, respectively (Figure 5A). Immunostaining images showed varied but largely co-localized patterns of Nestin+ and Ki67+ cells in the cultures (Figure 5B). Nestin+ cells were primarily concentrated in high cell-density aggregates that emerged within the monolayers. The co-localization of Nestin and Ki67 was most pronounced in the cell aggregates. These data suggest that a sub-population of Nestin+/Ki67+ cells could comprise a tumor stem cell niche that proliferated over time during ependymoma growth. 

### 3.6. Nestin+ Cell Enrichment by ECM in 3D Ependymoma Models 

To examine the effect of the ECM presence in 3D ependymoma cultures, we focused on Netin+ populations in 3D models. We compared silk scaffold-only models (3D-SF) and scaffolds infused with a collagen matrix (3D-SF/Col) or Matrigel models (3D-SF/Matri) (Figure 6). In the non-optimal media (e.g., N or N+FBS), Matrigel’s presence significantly increased Nestin+ cell percentage compared to scaffold-only models (Figure 6A(a)). For example, in the N media, the Matrigel-boosted Nestin+ population went from 9.5% in 3D-SF to 58% in 3D-SF/Matri (Figure 6A(b,c)).

When we combined the ECM types and media types for comparison, we found that the ECM effect superseded that of media types for 3D models (Figure 6B). In this study, EPN-4 samples were used. Regardless of the media type, e.g., N+E, DMEM+Supp., or N+E (no V), the presence of either the collagen gel or Matrigel significantly improved Nestin+ cell enrichment with the Matrigel showing a bigger effect (Figure 6B(a)). For example, in N+E (noV) media, the Nestin+ population increased from 25.9% in 3D-SF to 46.9% in 3D-SF/Col and 66.6% in 3D-SF/Matri (Figure 6B(b–d)). These data demonstrate that the ECM presence in 3D could further enrich Nestin+ tumor stem cells in ependymoma. 

### 3.7. 3D Tumor-Endothelial Co-Culture Model for Ependymoma

To recapitulate ependymoma and endothelial cell interactions, we generated 3D co-cultures by mixing endothelial cells into the ECM during gel infusion. The endothelial cells were initially introduced into the center hole region and were not in direct contact with scaffold-bound tumor cells. 

Con-focal images of the 3D models showed distinctive morphological changes in different 3D formats (Figure 7). The initial cell seeding onto the 3D scaffold resulted in single-cell attachment to the scaffold surfaces (Figure 7a). The tumor cells replicated and expanded into spheroids, which were anchored to the pores of the scaffold. These tumor spheroids consisted of Nestin+ and Ki67+ cells, and the tumor stem cell niche (Figure 7b,c). After the infusion of ECM gels into the scaffold, the tumor cells spread and migrated to fill the pores with dense and extensive Nestin+ and GFAP+ processes (Figure 7d,e). 

Upon endothelial cell introduction, after 2–3 weeks, PECAM+ endothelial capillary-like tubes formed (Figure 7f,g). The microvessels were found to connect with tumor cell aggregates in the scaffold region of the 3D model (Figure 7e), whereas in the center gel-filled region, they developed into a dense network (Figure 7f). In contrast, such tubular structures were not found in control 3D-SF/Col or 3D-SF/Matri models containing only endothelial cells. 

### 3.8. Transcriptomic Profiles of Ependymoma Cultures in Comparison with the Original Tumors 

To understand the transcriptomic alterations caused by the different in vitro culture conditions, we performed the RNA sequencing of primary ependymoma tissue and patient-derived in vitro cultures (Figure 8). We previously reported that the number of differentially expressed genes (adj. *p* < 0.05) between in vitro versus primary tissues was ~20% of ependymoma [42]. Here, we performed more a detailed analysis by comparing different culture conditions of ependymoma, focusing on two patients (EPN1 and EPN3). 

For this analysis, we also included endothelial cell cultures (*n* = 4, varying by 2D vs. 3D, cell density, and culture duration) to compare with the 3D tumor-endothelial co-cultures. For EPN1, the in vitro cultures included 3D-ECM (*n* = 3, varying by collagen or Matri gel) and the 3D-coculture (“3D-Co”, *n* = 8 varying by ECM type, culture media, and duration as described above). EPN3 did not have the 3D-ECM group, but had 3D-Co (*n* = 4, varying by ECM type, culture media), EPN “Sphere” cultures (e.g., tumor cell aggregates in suspension, *n* = 2, varying by media type), a 2D plate culture (*n* = 3, varying by media type) and 3D-SF cultures (*n* = 4 varying by media type). To examine VEGF’s influence, we also characterized the culture samples into with (“wV”) or without (“noV”) groups, depending on the initial culture media (for example, if a 3D-Co sample was cultured in a medium without VEGF but only with VEGF after coculture, the sample was classified as “noV”). 

Figure 8A shows a heatmap with unsupervised sample clustering. The heatmap shows the two primary tumor samples clustered closest together compared to in vitro cultures, which were separated by the patient regardless of the culture conditions. EPN3 cultures showed more similar expression profiles to the primary tissue than EPN1. Among EPN3 cultures, VEGF was found to play a critical role as samples with VEGF showed closer similarities to the primary tissue than without VEGF. 

Figure 8B shows the principal component analysis (PCA) of the expression data. The PCA plot showed the close clustering of EPN3 in vitro samples with the primary tumor tissue; however, it also showed clear separation from EPN1 in vitro samples. EPN2 in vitro samples overlapped with the primary tumor tissue on the PC1 component and only varied significantly on the PC2 component. Interestingly, EPN1 in vitro samples were clustered closely with the Endo samples, suggesting endothelial cell dominance in this group, which may have skewed the transcriptomic profiles away from the primary tumor tissue. 

Differential gene expression (DEG) analyses showed genes of significant differences between the paired groups, as visualized in the volcano plots in Figure 9. Compared to the primary tissue, the 3D-SF (Figure 9A) and Sphere (Figure 9B) groups both showed the upregulation of homeobox (HOX) genes, ART4 (ADP-ribosyltransferase 4) and COL13 (collagen alpha-1, XIII chain) genes. The 3D-Co group (Figure 9C), compared to the primary tissue, showed the upregulation of matrix metalloproteinases (MMPs), FGG (the gamma component of fibrinogen), and how, like the 3D-SF group, the TCF21 gene was a mesoderm-specific transcription factor. All three groups (3D-SF, Sphere, 3D-Co) showed the down-regulation of ODAD1: a gene associated with ciliary dyskinesia.

When compared between different 3D culture groups, fewer genes showed expression differences compared with the primary tumor tissue. Interestingly, both 3D-ECM (Figure 9D) and 3D-Co (Figure 9E) groups compared to 3D-SF showed the down-regulation of HOA genes compared to 3D-SF. The Sphere group (Figure 9F) compared to 3D-SF showed the up-regulation of ribosomal genes, which were associated with posterior fossa EPN relapses [15], and the down-regulation of genes associated with transport and secretion (e.g., SLC1A5, STC2, VIPR1), ECM remodeling (e.g., EFEMP1 and MMP3), lipid metabolism (e.g., ACSM3 and OSBPL10), tissue patterning (e.g., GREM2 and HOXA10), and mesoderm-specific transcription factors (e.g., TCF21 and TCF15). 

## 4. Discussion

Despite the need for ependymoma preclinical models, ependymoma in vitro models were lacking, and ependymoma cell lines were absent. Here, we developed in vitro ependymoma models with reconstituted fresh tumor cells from pediatric patient tumor tissues. We identified the equal portion mixture of the neural stem cell and endothelial cell culture media as a better liquid environment to support ependymoma cells’ in vitro growth compared to the conventional serum-containing culture medium. The ECM inclusion of collagen or Matrigel into the 3D scaffold further enhanced tumor in vitro growth, whereas Matrigel showed a greater pro-growth effect. Nestin+ cells appeared to form the tumor stem cell niche that predominated ependymoma’s in vitro growth. In addition, 3D co-cultures with endothelial cells promoted capillary formation around tumor cell aggregates, resembling the hallmark feature of the microvascular rosette in ependymoma. The transcriptomic analysis further revealed the critical role of VEGF in preserving tumor gene signatures. The tunable model format provided a basis for understanding the role of tumor microenvironmental regulators, now aided by the genetic information from this study’s transcriptomic analysis.

The tunable 3D silk protein-based scaffold for cell cultures allowed us to define the roles of specific microenvironmental factors, such as soluble factors, 2D versus 3D substrates, extracellular matrix, as well as their combinations for the optimal 3D in vitro growth of primary brain tumor cells [42]. Regarding these soluble factors, we found that the serum was harmful by inducing fibroblast-like cell overgrowth, suppressing Nestin+ tumor cells, and 3D culture viability. This finding was consistent with our earlier finding of the decreased metabolism of ependymoma cultures in FBS-containing culture conditions [35]. Growth differences in different media demonstrated the essential roles of the NeuroBasal background and VEGF since control cultures, i.e., with a DMEM base or without VEGF, had worse outcomes. The NeuroBasal base was originally optimized for cortical neuronal cultures, specifically with hypotonic osmolarity and abundant antioxidants [55]; its pro-growth effects, compared to DMEM, suggest the neuronal-like microenvironmental needs of ependymoma cells. FGF and EGF supplements in our improved media were also demonstrated to be potent pro-oncogenic factors in ependymoma by autocrine or paracrine actions [56,57,58,59].

Regarding the role of the 3D microenvironment, we found that different formats (e.g., 3D-SF, 3D-ECM, 3D-Co) changed tumor cell gene expression patterns. Homeobox genes are mostly affected in the 3D in vitro models compared to the primary tumor tissue and between different 3D formats. The homeobox genes encode a highly conserved family of transcription factors that play an important role in morphogenesis in all multicellular organisms. Interestingly, the homeobox genes are predominantly expressed in spinal ependymoma compared to intracranial ependymoma [60]. Whether the differential expression of the homeobox genes indicates a spatial tissue patterning cue from the artificial scaffold warrants further study. Nevertheless, this finding may provide an exciting new opportunity for future modulating highly conserved genes and pathways with bioengineering approaches.

Our study provides the first direct evidence of VEGF in promoting ependymoma growth and preserving tumor gene expression patterns. In the same defined media background, cultures with VEGF had better growth and Nestin cell enrichment than those without VEGF. Since our tumor cultures did not show the presence of endothelial cells (i.e., negligible PECAM+ population measured by flow cytometry), the cells’ better growth compared to those in media without VEGF suggested the direct role of VEGF in supporting ependymoma. While it has been reported that VEGF expression has no strong association with aberrant vascular patterns or angiogenic activity [43], other studies observed the significant co-expression of VEGF and Nestin in ependymomas of different grades and locations [46]. Our finding of VEGF-associated Nestin+ cell enrichment suggested that VEGF could augment cancer stem cell survival in ependymoma. In addition, the transcriptomic analysis showed that the close clustering of in vitro samples with primary tumor tissue was enabled only in conditions with a VEGF presence throughout the in vitro culture. Together, our results demonstrate that VEGF is essential in preserving both the phenotype and genotype of ependymoma in vitro.

Our in vitro ependymoma cultures demonstrated, for the first time, Nestin+ cell replication in the tumor growth niche, thus suggesting its role as the cancer stem cell of ependymoma. Nestin, an intermediate filament characteristic of neural progenitors, is commonly found in ependymoma [59]; however, the identity of the cell of origin of ependymoma is still unclear. Ependymoma is thought to derive from radial glia cells (RGCs) [60]: the embryonic neural progenitor cells for the terminally differentiated ependymal cells lining the spinal cord central canal and brain ventricles [61]. RGCs with a genetically engineered mutation can develop into ependymoma-like tumors in vivo [62]. These putative tumor cells-of-origin maintain neural stem/progenitor cell characteristics but with an abnormal ability to propagate and differentiate [34,63]. Classic ependymomas exhibit ependymal rosettes that mimic the normal ependymal canal [60]. In our 2D cultures, similar rosette-looking patterns emerged from confluent monolayers; Nestin and Ki67 double-positivity suggested a cancer stem cell-enriched growth niche. In the 3D silk fibroin-based scaffolds, individual tumor cells developed Nestin and Ki67 double-positive tumor spheroids. By charting the time-evolved expansion of Nestin+/Ki67+ cells as well as their spatial location, our in vitro models provided a platform for examining the abnormal transformation and progression of ependymoma-associated stem/progenitor cells.

The most interesting finding was the capillary network developed in 3D ependymoma-endothelial co-cultures, presenting in vivo-like tumor vasculature. The capillary network did not form in 3D endothelial cultures alone, suggesting that the tumor cells provided key signals for endothelial remodeling. Conversely, the presence of endothelial cells shifted the gene expression patterns of the 3D tumor models, especially of genes associated with ECM components and remodeling. Interestingly, the predominance of 3D co-cultures in EPN2 in vitro samples could have contributed to the separation of EPN2 in vitro models from the primary tumor tissue regarding their transcriptomic profiles. Future studies need to control for the relative ratio of tumor and endothelial cells in 3D co-culture models to better elucidate the role of endothelial cells in ependymoma vasculature.

## 5. Conclusions

We established human patient-derived in vitro ependymoma models for recapitulating the tumor’s in vivo phenotypic and transcriptomic features. Our study provided chemically defined media and culture conditions that led to the enrichment of Nestin+ tumor stem/progenitor cells and the formation of a tumor growth niche in both 2D and 3D. The 3D tumor-endothelial co-culture model demonstrated, for the first time, an in vivo-like ependymoma vasculature showing a dynamic interplay between the tumor and the tumor vasculature with implications for gene expressions. This tunable model format provides a basis for examining other microenvironmental regulators in tumor growth, now aided by the genetic information from this study’s transcriptomic analysis.

## Figures and Tables

**Figure 1 bioengineering-10-00840-f001:**
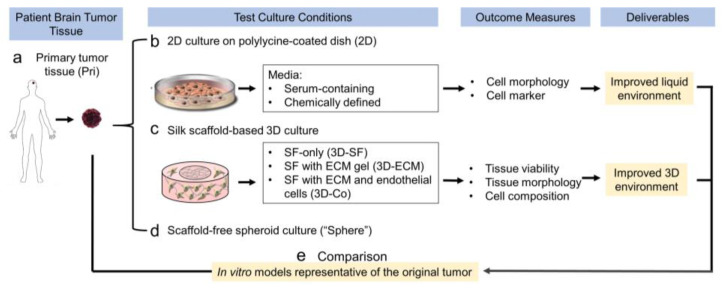
Design of ependymoma in vitro models.

**Figure 2 bioengineering-10-00840-f002:**
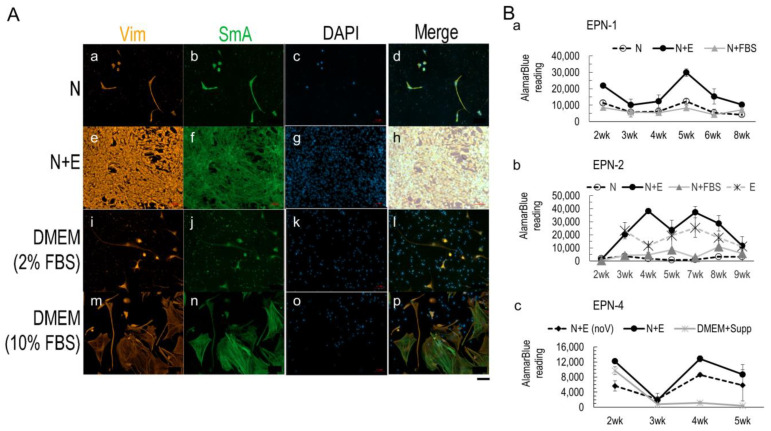
Media conditions affecting 2D and 3D culture viability. (**A**) Representative fluorescence images of EPN-1 2D cultures in different media, N (a–d), N+E (e–h), DMEM supplemented with 2% (i–l) and 10% FBS (m–p). Cultures of 3 weeks were stained for vimentin (Vim, red), smooth muscle actin alpha (SmA, green), and nuclei DAPI stain (blue). Scale bar, 100 µm. (**B**) AlamarBlue assay for the viability of 3D cultures of EPN-1 (a), EPN-2 (b), and EPN-4 (c). Error bar, standard error of the mean. N = 4/group.

**Figure 3 bioengineering-10-00840-f003:**
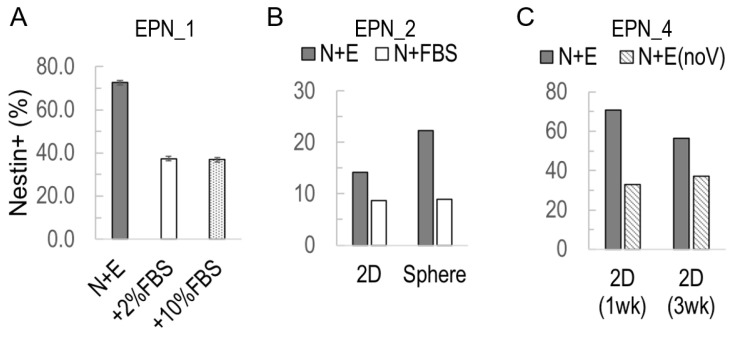
Nestin+ cell enrichment in 2D ependymoma cultures. Nestin+ cell percentage was measured by flow cytometry. (**A**) EPN-1 tumor cells in serum-free N+E media compared to those with 2% or 10% FBS. (**B**) EPN-2 tumor cells in serum-free N+E media compared to those with 10% FBS and also as 2D cultures versus in suspension (sphere). (**C**). EPN-4 tumor cells in N+E media compared to those in VEGF-missing N+E media “N+E (no V)”.

**Figure 4 bioengineering-10-00840-f004:**
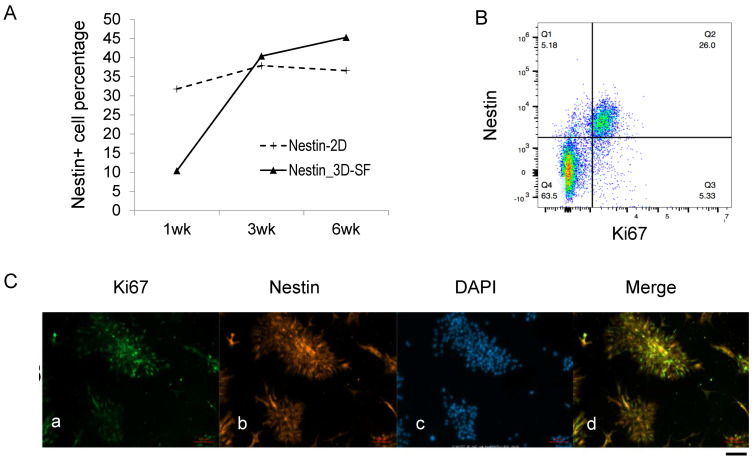
Nestin+ and Ki67+ colocalization. (**A**) Nestin+ cell population change throughout culture duration in 2D versus 3D cultures. (**B**) Nestin+ and Ki67+ double positive population in EPN-3 3D cultures. The pseudocolor of the dots denotes areas of high (red), medium (green) or low (blue) population density. Axes show log-scale fluorescence intensity by flow cytometry. Nestin, y-axis; Ki67, x-axis. (**C**) Representative fluorescence images of EPN-3 2D cultures stained with Ki67 (a), Nestin (b), DAPI, and (c) All three markers merged (d). Note the rosette-looking morphology. Scale bar, 100 µm. All data are from EPN-3 cultures.

**Figure 5 bioengineering-10-00840-f005:**
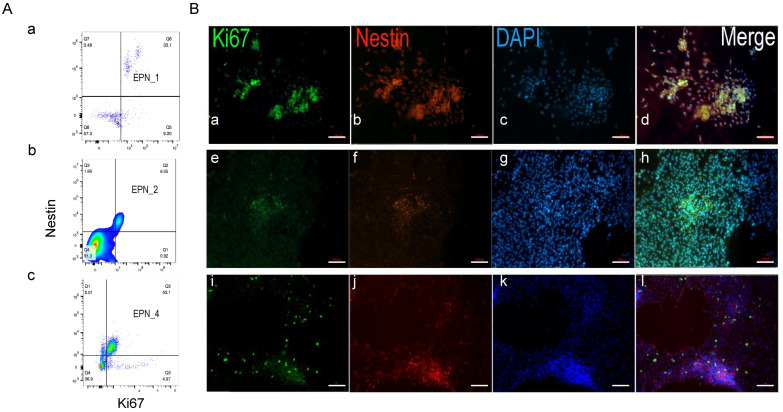
Nestin+ and Ki67+ colocalization in the other three ependymoma patient-derived cultures. (**A**) Nestin+ and Ki67+ double positive populations in 3D cultures (a), EPN-1; (b), EPN-2; (c), EPN-4. The pseudocolor of the dots denotes areas of high (red), medium (green) or low (blue) population density. Axes show log-scale fluorescence intensity by flow cytometry. Nestin, y-axis; Ki67, x-axis. (**B**) Representative fluorescence images of ependymoma 2D cultures stained with Ki67 (green), Nestin (red), DAPI (blue), and all three markers merged, EPN-1 (a–d), EPN-2 (e–h), EPN-4 (i–l). Note the high cell-density clusters compared to the surrounding monolayers. Scale bar, 100 µm.

**Figure 6 bioengineering-10-00840-f006:**
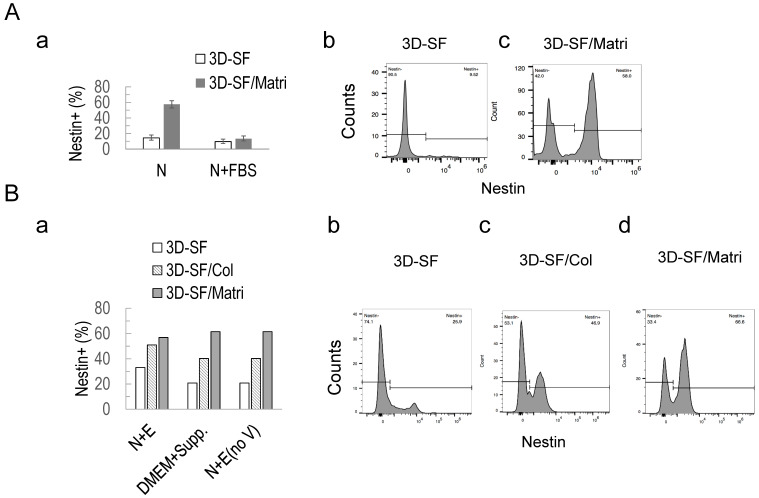
Nestin+ cell enrichment in 3D ependymoma cultures. (**A**) (a) Nestin+ cell population in non-optimal media, N and N+FBS of EPN-1 3D cultures of scaffold-only (3D-SF) compared to scaffold infused with Matrigel (3D-SF/Matri). Histogram of Nestin+ cell population in N media showing a pronounced increase in 3D-SF/Matri (c) compared to 3D-SF (b). (**B**) (a) Nestin+ cell population in three media conditions: N+E, DMEM plus supplements, and N+E without the VEGF (no V) of EPN-4 3D cultures of scaffold-only (3D-SF) compared to scaffold infused with collagen (3D-SF/Col) or Matrigel (3D-SF/Matri). Histogram of Nestin+ cell population in N+E (no V) media showing significant differences in different 3D formats, 3D-SF (b), 3D-SF/Col (c), and 3D-SF/Matri (d).

**Figure 7 bioengineering-10-00840-f007:**
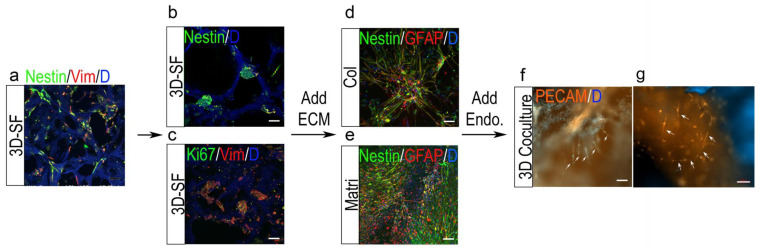
The 3D ependymoma-endothelial co-cultures produced capillary-like networks. Dissociated tumor cells were initially seeded relatively sparsely onto the 3D scaffold surfaces, as shown with Nestin (green) and Vimentin (Vim, red) staining (**a**). Tumor spheroids developed within the scaffold pores by 3 weeks in the culture showing Nestin+ ((**b**), green) and Ki67+ ((**c**), green) tumor cells. Upon ECM gel infusion, 3D cultures showed an increased cell density and extension of the cell processes of Nestin (green) or GFAP (red) positive staining throughout scaffold pores ((**d**), Col; (**e**), Matri). In 3D co-cultures with endothelial cells, capillary-like tubes (arrows) with positive PECAM staining (orange) were seen in gel-infused areas in the 3D models of the scaffold region (**f**) and the center gel-filled region (**g**). Note that the scaffold material autofluoresces in purplish blue (**a**–**c**,**g**) Scale bar, 100 µm.

**Figure 8 bioengineering-10-00840-f008:**
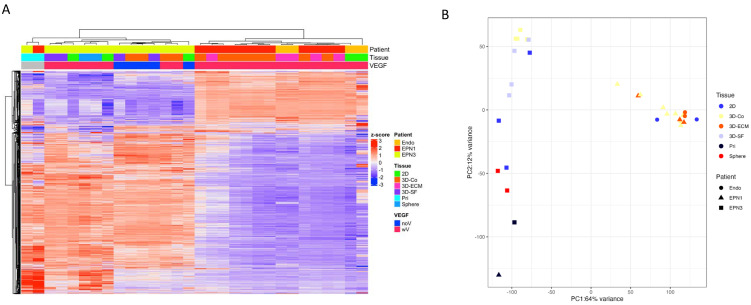
Transcriptomic analysis of ependymoma primary tissue and in vitro cultures. (**A**) Heatmap of z-scores from gene counts of EPN primary tumor tissue (*n* = 2, EPN-1 and -3) and patient-derived in vitro cultures (*n* = 11 for EPN-1, *n* = 13 from EPN-3) and endothelial cultures (*n* = 4). The dendrogram from hierarchical clustering is shown on the top with sample annotation (“Patient”, “Tissue”, and “VEGF”). The sample annotation bar shows corresponding colors. (**B**) Principal component analysis plot of the first two PCs from the top 5000 genes by variance. “Patient” groups are marked by dot shapes, and “Tissue” types by dot colors.

**Figure 9 bioengineering-10-00840-f009:**
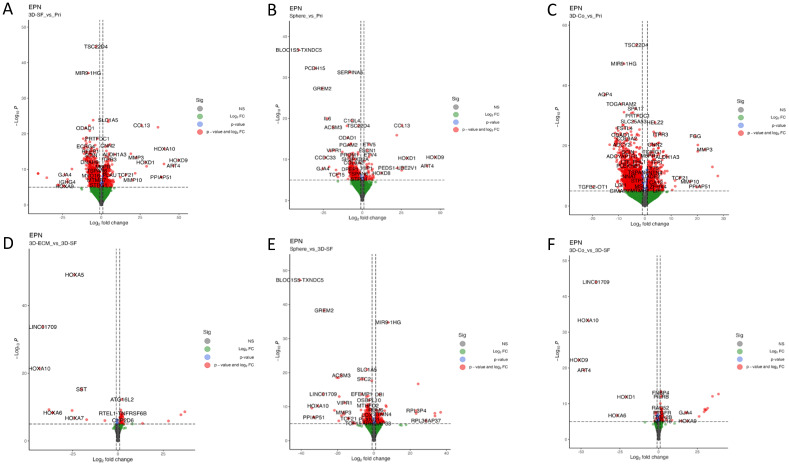
Volcano plots of differential gene analysis. (**A**) 3D-SF versus the primary tumor tissue (Pri). (**B**) Sphere versus Pri. (**C**) 3D-Co versus Pri. (**D**) 3D-ECM versus 3D-SF. (**E**) 3D-Co versus 3D-SF. (**F**) Sphere versus 3D-SF.

**Table 1 bioengineering-10-00840-t001:** Ependymoma Models in Literature.

	Reference	Sample Size	Approach	Note
2D primary culture	Pertuiset et al., 1985 [23]	2	Tumor biopsies in 2D culture	The mean tumor cell doubling time was 46 h for two ependymomas.
	Engebraaten et al., 1990 [24]	1	Tumor biopsies cut into fragments of 0.5 mm diameter and placed in agar overlay tissue culture	Co-culture with fetal rat brain tissue to examine tumor invasiveness
	Jennings et al., 1994 [25]	2	Tumor biopsies within passage 1–4	TGF beta 1 and TGF beta 2 are potential growth regulators for ependymomas
	Ishiwata et al., 2004 [26]	2	Tumor biopsies in 2D culture	The ependymoma cells formed a rosette-like cell arrangement. Primary and long-term (>3 mon) cultures were established, but not cell lines.
Cell line development	Nakagawa et al., 1983 [27]	1	2D with Eagle’s MEM plus L-15 medium (6% fetal calf serum)	Grew like a sheet in groups. In soft agar, formed solid tumor differed from the original ependymoma.
	Hussein et al., 2011 [28]	2	Monolayers were grown in tumor medium: DMEM/L-glutamine supplemented with 15% FBS. To generate neurospheres, cells grown as monolayers were washed, dissociated and resuspended into the serum-free stem cell medium: DMEM high glucose and Ham’s F-12 solution (70/30%), 2% B27, 5 ng/mL heparin, supplemented with 20 ng/mL human recombinant epidermal growth factor (hrEGF;), and 20 ng/mL human basic recombinant fibroblast growth factor (bFGF).	Two cell lines were established. Monolayers were passaged for >60 generations. Neurospheres were serially passaged for up to 11 generations.
	Sanden et al., 2015 [29]	1	UltraCULTURE™ cell culturing medium supplemented with 2 mM L-glutamine,1% Penicillin-Streptomycin, bFGF (40 ng/mL), and EGF (20 ng/mL) every 3–4 days.	Monolayers for >passage 5, but not 3D spheroids
	Amani et al., 2017 [19]	2	Cells were plated in Optimem media supplemented with 15% fetal bovine serum and cultured in either (1) ultra-low attachment plates to form nonadherent cultures or (2) using standard tissue culture treated plates to generate adherent monolayer cultures.	Two unique cell lines of intracranial, posterior fossa 1q+ ependymoma. The success in establishing these lines potentially stems the fact that the cells were from recurrent and collected from sites of intracranial metastasis rather than primary tumor. In contrast, the laboratory has attempted to establish cell lines from first occurrence ependymoma cases for almost 20 years (∼50 cases), none of which yielded a stable cell line.
	Pavon et al., 2018 [30]	5	The isolated cells were cultured in Dulbecco’s Modified Eagle’s Medium-Low Glucose (DMEM-LG) supplemented with 10% Fetal Bovine Serum and antibiotics.	Primary cell cultures were successfully obtained from five tumor samples. The success rate of isolating EPN cell cultures from all samples was around 70%. GFAP/CD133+CD90+/CD44+ ependymoma cells maintained key histopathological and growth characteristics of the original patient tumor.
	Yuan et al., 2021 [31]	2	Cells were co-cultured with irradiated 3T3 fibroblasts in the presence of Rho kinase (ROCK) inhibitor Y-27632.	Conditional reprogramming resulted in robust increases in growth for a majority of these tumors, with fibroblast conditioned media and ROCK inhibition both required. Cells were stable for up to 27 passages in terms of their appearance, with a doubling time of approximately 30 h after 8 passages
3D neurosphere	Yamada et al., 2002 [32]	2	The tumors were minced, and small fragments were prepared and embedded in the collagen gel	Ultrastructural observations. A basement membrane was formed surrounding the tumor cell processes facing the collagen gel in two ependymomas.
	Brisson et al., 2002 [33]		3D co-culture with endothelial cells in Matrigel	The morphological features (microvilli, cilia, and caveolae) of these cultured cells were similar to those of the tumor in vivo.
	Thirant et al., 2011 [34]	10	Cells from fresh or cryo-frozen biopsies were cultured in NSA-H medium with 10 ng/mL FGF, 20 ng/mL EGF, and 1 mg/mL Heparin. The cells were further cultured until appearance of floating cellular spheres.	Limited self-renewal, stopped proliferating within 5 months followed by progressive disappearance. 3 cases no renewal. 3 cases renewal > 7 times, 4 cases < 7 renewal times.
3D scaffold model	Sood D., et al., 2019 [35]	1	3D silk protein-based scaffold and collagen and Matrigel infusion	The 3D brain ECM-containing microenvironment supports distinctive phenotypes associated with tumor type-specific and ECM-dependent patterns in the tumor cells’ transcriptomic and release profiles.
Transgenic mouse model	Johnson et al., 2010 [36]	1	A mouse model by selecting neuronal stem cells with a deleted Ink4a/Arf locus that overexpress tyrosine receptor ephrin (EphB2)	Cross species genomics matches driver mutations and cell compartments to model ependymoma.
Patient-derived xenograft (PDX) or orthotopic xenograft (PDOX)	Yu et al., 2010 [37]	1	Transplanting a fresh surgical EPN tissue from a pediatric patient into the brain of immune deficient mice	A clinically relevant orthotopic xenograft model of ependymoma that maintains the genomic signature of the primary tumor and preserves cancer stem cells in vivo.
	Pierce et al., 2019 [20]	2	Disaggregated tumors from 2 1q+ PFA patients were injected into the flanks of NSG mice.	Establishment of patient-derived orthotopic xenograft model of 1q+ posterior fossa group A ependymoma
	Whitehouse et al., 2023 [22]	1	The establishment of a patient-derived orthotopic xenograft (PDOX) model of posterior fossa A (PFA) EPN, derived from a metastatic cranial lesion.	Despite the aggressive nature of the tumor in the patient, this PDOX was unable to be maintained past two passages in vivo

**Table 2 bioengineering-10-00840-t002:** Ependymoma (EPN) Cases.

Case ID	Grade	Age	Sex	Location
EPN-1	II	15 months	Male	Posterior fossa
EPN-2	II	11 months	Male	Posterior fossa
EPN-3	III	2 years	Female	Posterior fossa
EPN-4	III	7 months	Male	Posterior fossa

## Data Availability

The RNAseq data presented in this study can be accessed from the Sequence Read Archive (SRA) on the NCBI website (SUB10713222).

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
