# Peer review of "Human Patient-Derived Brain Tumor Models to Recapitulate Ependymoma Tumor Vasculature"

_bioengineering, 2023, doi:10.3390/bioengineering10070840_

Round 1

Reviewer 1 Report

This work used a tunable 3D silk protein-based scaffold to approximate the 3D ependymoma microenvironment for reconstituting a patient's tumor in vitro. The data showed that mixed neural stem cell and endothelial cell media without serum supported in vitro growth of all four ependymoma cases tested. The paper's contribution to existing knowledge in this research field is justified. The paper needs to contribute more; the following points can improve the manuscript.

1.     Enhance the introduction to show the motivation for this work. One paragraph should be added to show this.

2.     A literature gap should be given. A comparative study can be added to a related work section in table form to show the recent efforts.

3.     Figure 1 caption can be shifted to the text. Keep it as “Figure 1. Design of ependymoma in vitro models.”

4.     Table 1 is an image; it is better to rewrite it as text, not an image.

5.     Figure 2 caption can be shifted to the text. Keep it as “Figure 2. Media conditions affect 2D and 3D culture viability.” It also needs to be improved.

6.     Figures 3 and 4 need to be clarified. They are overlapped.

7.     Figures 5, 6, 7, and 8 captions can be shifted to the text.

8.     References should be updated; there are no references in 2023.

9.     The manuscript organization should be improved. 

10.  Strong conclusion recommended.

11.  The paper is unsuitable for acceptance in its current form. The article needs rewriting to address the comments mentioned above. 

Minor editing of English language required.

Author Response

  1. Enhance the introduction to show the motivation for this work. One paragraph should be added to show this.

Reply: We added one paragraph (the last one in the Intro.) about the motivation of this work, as recommended.

  1. A literature gap should be given. A comparative study can be added to a related work section in table form to show the recent efforts.

Reply: We added Table 1 to summarize all the major literature in ependymoma modeling.

  1. Figure 1 caption can be shifted to the text. Keep it as “Figure 1. Design of ependymoma in vitro models.”

Reply: We moved Figure 1 caption to the text (p7, line 235-243).

  1. Table 1 is an image; it is better to rewrite it as text, not an image.

Reply: We re-formatted Table II (old Table I) accordingly.

  1. Figure 2 caption can be shifted to the text. Keep it as “Figure 2. Media conditions affect 2D and 3D culture viability.” It also needs to be improved.

Reply: We respectively disagree, since details need to be provided in figure legends as some readers prefer to read figures but not the main text. We made improvement of the new Figure legend.

  1. Figures 3 and 4 need to be clarified. They are overlapped.

Reply: We re-formatted Figure 3 and 4 for better clarification.  

  1. Figures 5, 6, 7, and 8 captions can be shifted to the text.

Reply: See above reply to 5.  

  1. References should be updated; there are no references in 2023.

Reply: We revised the reference list, and added most recent ones in 2023 (e.g. Ref. 37).  

  1. The manuscript organization should be improved. 

Reply: We made major re-organization in all sections of the manuscript.  

  1. Strong conclusion recommended.

Reply: We strengthened our Result and Conclusion (highlighted in yellow).  

  1. The paper is unsuitable for acceptance in its current form. The article needs rewriting to address the comments mentioned above. 

Reply: See above replies.  

Reviewer 2 Report

In the manuscript entitled “Human Patient-derived Brain Tumor Models to Recapitulate Childhood Ependymoma and Tumor Vasculature”, Tang-Schomer et al. developed an in vitro model of ependymoma tumor using reconstituted fresh tumor cells from pediatric patient tumor tissue. The authors demonstrated that an equal portion mixture of neural stem cell and endothelial cell culture media provided a better liquid environment to support the ependymoma cells’ growth in vitro. Moreover, they showed that 3D co-cultures with endothelial cells promoted capillary formation, which behaved similarly to a microvascular rosette in ependymoma. This study is based on a tunable 3D silk protein scaffold, which was developed and previously published by the same authors. Compared to their previous studies, this work presents a new attempt to optimize patient-derived in vitro ependymoma models for in vivo mimicking. Overall, this paper is well-written, and the methods are well-documented. The manuscript can be considered for publication, provided the concerns listed below are addressed.

1.     Is the word “and” in the title necessary? Should the phrase simply be “Ependymoma Tumor Vasculature”?

2.     All abbreviations used in the abstract, e.g., VEGF, ECM, should be described with their full names.

3.     The y-axis labels in Figure 2B are wrong. They are supposed to be the brightness/readout of the images. However, the authors labeled them as “Ex/Em 560/590”, which is misleading. The correct labels should be used.

4.     There are major problems in the format of this manuscript. Figure 3 is largely overlayed by Figure 4.

5.     The images in Figure 4C seem to be overlaying some other images. Please double-check the formatting before resubmission.

6.     The y-axis label in Figure 4A is confusing. What is the “percentage”. It should be more specific.

7.     What are the units of the x- and y-axes in Figures 4B, 5A?

8.     The scale bars are very hard to see in some figures, e.g., Figure 5B. Please make them evident.

Author Response

  1. Is the word “and” in the title necessary? Should the phrase simply be “Ependymoma Tumor Vasculature”?

Reply: We modified the Title accordingly.

  1. All abbreviations used in the abstract, e.g., VEGF, ECM, should be described with their full names.

Reply: We provided full names of the abbreviations in the revised abstract.  

  1. The y-axis labels in Figure 2B are wrong. They are supposed to be the brightness/readout of the images. However, the authors labeled them as “Ex/Em 560/590”, which is misleading. The correct labels should be used.

Reply: We revised Figure 2B y-axis label accordingly.  

  1. There are major problems in the format of this manuscript. Figure 3 is largely overlayed by Figure 4.

Reply: We re-formatted Figure 3 and 4 for better clarification.  

  1. The images in Figure 4C seem to be overlaying some other images. Please double-check the formatting before resubmission.

Reply: We reformatted Figure 4 for better clarification.  

  1. The y-axis label in Figure 4A is confusing. What is the “percentage”. It should be more specific.

Reply: We revised Figure 4A y-axis label accordingly.  

  1. What are the units of the x- and y-axes in Figures 4B, 5A?

Reply: The units are fluorescence intensity in log scale, as standard of flow cytometry scatter plots (Figures 4B, 5A). We clarified the units in the revised figure legends (highlighted in yellow).

  1. The scale bars are very hard to see in some figures, e.g., Figure 5B. Please make them evident.

Reply: We revised the scale bars in Figure 5B and make them more evident.

Reviewer 3 Report

The paper submitted by the authors is quite interesting and provides important information in this field. Nonetheless, some sections (mainly material and methods, and results) should be revised since they could seem unclear.   In the abstract, there are different abbreviations that should be explained before. In addition, I recommend to use these abbreviations into the body text but not here. Line 240, please explain the abbreviation. In general, the results section is mixed with some statements that should be in material and methods. In addition, it is quite uncommon to include references in this section. Lines 258-268, the entire paragraph doesn’t fit here. It should be moved to discussion section. The conclusion section should be improved showing the more relevant findings of your study.

Author Response

Reply: We provided full names of the abbreviations in the revised abstract.  

The Results section has been re-written to better align with the Material and Methods.

Some literature is still included in the Result section to provide context of the specific marker use (e.g., Vim) or gene functions (e.g. new Figure 9 related text).

The old Lines 258-268 have been deleted.

We improved our Result and Conclusion to show the more relevant findings of our work (highlighted in yellow).  

Round 2

Reviewer 1 Report

The authors have addressed most of my concerns. The paper can be accepted. 

Reviewer 3 Report

The authors have corrected the paper and can be accepted in the current form